# Benchmark for Automatic Clear-Cut Morphology Detection Methods Derived from Airborne Lidar Data

**Zlatica Melichová** [1,*], **Stano Pekár** [2] and **Peter Surový** [1]

1 Department of Forest Management, Faculty of Forestry and Wood Sciences, Czech University of Life Sciences Prague, Kamýcká 129, 16500 Prague, Czech Republic

2 Department of Botany and Zoology, Faculty of Science, Masaryk University, Kotlářská 2, 61137 Brno, Czech Republic

* Correspondence: melichovaz@fld.czu.cz

**Abstract:** Forest harvest detection techniques have recently gained increased attention due to the varied results they provide. Correctly determining the acreage of clear-cut areas is crucial for carbon sequestration. Detecting clear-cut areas using airborne laser scanning (ALS) could be an accurate method for determining the extent of clear-cut areas and their subsequent map display in forest management plans. The shapes of ALS-detected clear-cut areas have uneven edges with protrusions that might not be readable when displayed correctly. Therefore, it is necessary to simplify these shapes for better comprehension. To simplify the shapes of ALS-scanned clear-cut areas, we tested four simplification algorithms using ArcGIS Pro 3.0.0 software: the retain critical points (Douglas–Peucker), retain critical bends (Wang–Müller), retain weighted effective areas (Zhou–Jones), and retain effective areas (Visvalingam–Whyatt) algorithms. Ground-truth data were obtained from clear-cut areas plotted in the forest management plan. Results showed that the Wang–Müller algorithm was the best of the four ALS algorithms at simplifying the shapes of detected clear-cut areas. Using the simplification algorithm reduced the time required to edit polygons to less than 1% of the time required for manual delineation.

**Keywords:** multitemporal laser scanning data; harvest detection; simplification polygons; clear-cut areas

## 1. Introduction

In Europe, sustainable forest management has become a prominent topic, as management decisions impact forest growth, composition, and structure, as well as wood production, carbon sequestration, and nature conservation in both temporal and spatial contexts [1]. Forestry has been recognized as an important way to reduce $CO_2$ emissions and combat global warming, as highlighted in the Paris Agreement [2]. Nevertheless, forestry harvesting practices may adversely affect the benefits of forest $CO_2$ capture. Research comparing the impact of reducing carbon sequestration through harvesting versus natural disturbances has shown that harvesting has a greater impact. Notably, incidental harvesting, in which the effects of harvesting cannot be easily distinguished from those of natural disturbances, must also be considered [3].

Various techniques and methods for detecting harvests and clear-cuts deserve attention, as their accurate evaluation is crucial. Incorrect use of techniques or reference data can result in erroneous conclusions. Ceccherini et al. (2020) [4] focused on harvest detection in Europe using satellite data. They tracked the increase in harvested forests and biomass losses for 2016–2018 and compared them to those for 2011–2015. Their findings indicated that harvesting had increased by 34% on average, potentially impacting biodiversity, soil erosion, and water regulation. Their study suggested that the expansion of the timber market, wood-based bioenergy, and international trade had led to an increase in harvesting speed. The authors warned that continued high harvesting rates could impede forest-based

efforts to mitigate climate change impacts. Picard et al. (2021) [5] investigated claims regarding increased harvesting in European countries and re-examined France using original data reported in [4]. They found that the rate of change in harvested area depended on the comparison period used, and that data regarding extraction volumes from different sources produced varied results. Responding to an investigation in Finland and Sweden, which found only marginal or no harvesting increase after 2015 [6], the authors suggested that sensitivity and detection of harvesting areas and overburden were increasing [4]. Ceccherini et al.'s (2020) [4] study highlighted the potential for inaccurate results when inappropriate estimates and reference data are used in satellite data analysis.

Remote sensing technologies and three-dimensional data offer increasingly accurate options for estimating growth rates and forest information. Lidar, satellites, and unmanned aerial vehicles (UAVs) are remote sensing methods used for gap mapping, including size and spatial distribution parameters [7]. ALS technology enables 3D characterization of forest canopies, allowing digital terrain model (DTM) and digital surface model (DSM) calculations to describe treetops from the original point cloud [8]. An advantage of using lidar data is the resulting accuracy of DTMs, which are often available for public use [9]. Subtracting an area's digital surface model from its digital terrain model produces a canopy height model (CHM), which is widely used as the basis for various forestry analyses [10].

Forest metrics can be computed using lidar directly from the point cloud or rasterized point cloud data (a rasterized point cloud is a raster in which each cell is described by height value) [11]. Rasterized clouds are usually faster and easier to process [12]; however, they offer less information (metrics) than point clouds. Therefore, rasterized clouds are more suitable for clear-cut detection.

Two approaches are generally used to derive forest information: the area-based approach (ABA) and individual tree detection (IDT) [13]. These methods are typically used to estimate forest characteristics, stand-level biomass, volume, or basal area. For estimation purposes, these variables mostly use the plot-level method, which involves calculating various descriptive statistics, such as mean, maximum, standard deviation, and height metrics, counting height percentiles. These statistics can be used to characterize different aspects of the point cloud's structure, such as density or point distribution [14]. In comparing point clouds from different time steps, it is possible to measure changes in vegetation variables such as growth, increment, and site index [15].

An important aspect of map creation using lidar sources (either point clouds or rasterized clouds) is map readability and the balance between the amount of detail and readability. A higher level of detail includes more information, but such maps are not easily readable, which can lead to erroneous conclusions, as noted in [4]. Therefore, there is a need for standardized and objective simplification of map products (usually called feature generalization or simplification).

Generalizing a map involves simplifying, removing details from, enlarging, or modifying a map so that its final form is as legible and understandable as possible, while preserving source data and essential map attributes [16]. In this study, we evaluated four algorithms for polygon simplification created by the automatic subtraction of two consecutive lidar scans: the Douglas–Peucker, Visvalingam–Whyatt, Zhou–Jones, and Wang–Müller algorithms.

The Douglas–Peucker algorithm reduces the number of points in a curve that is approximated by a series of points, depending on the maximum distance between the original curve and the simplified curve. The algorithm recursively eliminates points that are closer to the line connecting the two endpoints of the curve than the specified tolerance [17]. This algorithm is also known as the Ramer–Douglas–Peucker algorithm, after Urs Ramer in addition to David Douglas and Thomas Peucker, who independently developed it in 1972 and 1973, respectively. The algorithm is widely used in computer graphics, cartography, and GIS applications. Despite the fact that the Douglas–Peucker algorithm was developed to simplify watercourse lines so that redundant points could be removed while preserving information, it also has applications in digital cartography [18]. It is a vertex

subsampling algorithm that independently alters individual polylines using a simplification process that considers only the sequence of vertices within the polyline itself, without considering surrounding features. The algorithm's output consists of a subsequence of the original polyline's vertices that represents the simplified output polyline's vertices in their original order [19]. The Douglas–Peucker algorithm and the Visvalingam–Whyatt algorithm are prone to remove small bends, resulting in less accurate representations of small watercourses [20]. Unlike these algorithms, the Wang–Müller algorithm preserves the characteristic properties of natural features [21]. In addition to line simplification, the Douglas–Peucker and Visvalingam–Whyatt algorithms are also suitable for line segmentation [21] or generalization [22]. The Visvalingam–Whyatt algorithm also preserves the geometry of an area while smoothing its contours [23].

To evaluate the accuracy of these algorithms, one can interpret their results as a raster classification result. The classification process is widely used to transform image data into map products; each pixel is categorized into one of multiple categories: typically two or more [24]. A confusion matrix is commonly used to describe thematic map accuracy and to compare accuracies. However, it can also be used to derive more useful information, such as refining estimates of the areal extent of classes in a region and optimizing a thematic map for a particular user. This can be accomplished by using the matrix together with information regarding actual error costs of the map's value. The reliability of the confusion matrix is important, as issues such as sample design and ground data accuracy can affect its accuracy [25].

In this study, we identified and estimated clear-cuts using ALS data and compared them to clear-cuts indicated on a forest management plan created by a human operator. Four polygon simplification algorithms were compared with each other, and their accuracies were assessed using the forest management plan, which served as ground-truth data. The automatic map creation method provides consistency and repeatability, and it can significantly decrease the time required for manual delineation.

## 2. Materials and Methods

### 2.1. Study Site

Our research area of interest was the School Forest Enterprise in Kostelec nad Černými lesy. ŠLP Kostelec nad Černými lesy is a university forestry estate of the Czech University of Agriculture in Prague. It is located 25–50 km southeast of Prague (Figure 1). The area's altitude varies from 210 to 528 m, its average annual temperature is 8.14 °C, and its average annual precipitation is 663 mm [26]. The area is approximately 6000 ha, and it is actively managed. The area is in beech–oak (21%), oak–beech (53.8%), and beech (25.2%) vegetation stages. Its tree species composition includes Norway spruce (*Picea abies* (L.) H. Karst.) (55%), Scots pine (*Pinus sylvestris* L.) (18%), European beech (*Fagus sylvatica* L.) (12%), Sessile oak (*Quercus petraea* (Matt.) Liebl.) (9%), European silver fir (*Abies alba* Mill.) (2%), hornbeam (*Carpinus betulus* L.) (1%), and other woods (3%). There are also several protected areas within the ŠLP territory, of which the Voděradské Bučiny National Nature Reserve, with an area of 683 ha, is one of the most important.

### 2.2. Data Processing

Data from airborne laser scanning (ALS) were processed in the Anaconda programming environment (Anaconda, Inc., Austin, TX, USA), using the Python programming language, version 3.11.1 (Python Software Foundation, Beaverton, OR, USA). The Point Data Abstraction Library (PDAL), version 2.6.0 (Hobu, Inc., Iowa City, IA, USA), a library equipped with prebuilt commands for various analyses, was employed to interpret laser data. With the help of this library, data were filtered, classified, and converted from laser point clouds to raster data; the PDAL was used to classify ground returns using the simple morphological filter (SMRF) technique, version 2.6.0 (Hobu, Inc., Iowa City, IA, USA). This algorithm effectively discriminated points into two distinct groups: ground and nonground. Initially, an outlier filter was applied to classify outliers with a classification value of 7.

These outliers were subsequently excluded during SMRF processing using the "ignore" option. Finally, a range filter was implemented to extract the ground returns, identified by a classification value of 2 [27].

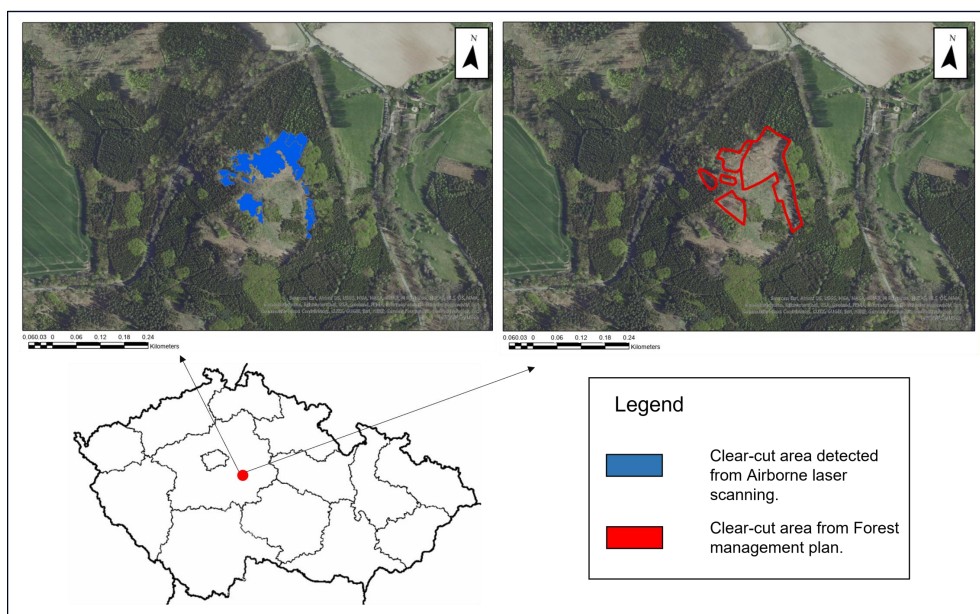

**Figure 1.** The study area.

Rasterization

We used the PDAL to create a raster surface utilizing a fully classified point cloud via PDAL's writers.gdal functionality [28]. ArcGIS Pro 3.0.0 software (Environmental Systems Research Institute (ESRI), Redlands, CA, USA) was used for subsequent data processing.

The readability problem originates in the actual shapes of detected clear-cut areas. Although the maps are geometrically and positionally correct, they appear strange to the human observer and may become illegible, for example, in larger scale contour maps. Thus, a simplification of their shape is inevitable.

Two processes were needed to simplify and smooth surfaces to solve the two problems visible in Figure 2. The first was to close the holes in the polygons. These holes represented individual standing trees around which the parent growth had already been removed. The second problem was the polygon's shape, which contained complicated curvatures. The "eliminate polygon part" tool was used to address the first problem; we used this tool to close holes created inside the polygons (Figure 2a).

A sample of polygons of clear-cut areas was selected for evaluation. The shapes of clear-cut areas in the forest management plan were used as validation data. Figure 2 shows a comparison of the shape of one clearing sample resulting from airborne laser scanning and the validated clear-cut area from the forest management plan.

In this study, we used multitemporal laser datasets from 2021, 2020, and 2019. We subtracted the 2021 and 2020 laser datasets from each other to form the 2021 clear-cut areas, and we used the same approach for the 2020 and 2019 laser datasets to form the 2020 clear-cut areas. In the first step, we removed the 2020 and 2021 ALS shape overlap using the "erase" tool and then combined them using the "merge" tool. We did not separately evaluate the clear-cut area for each year because the 2021 forest management plan plotted the clear-cut area and also included the 2020 clear-cut area. Clear-cut area classification results from the laser data contain holes (small polygons) that characterize individual trees or vegetation that are mapped at a greater resolution than the established threshold for classifying clear-cuts (Figure 3).

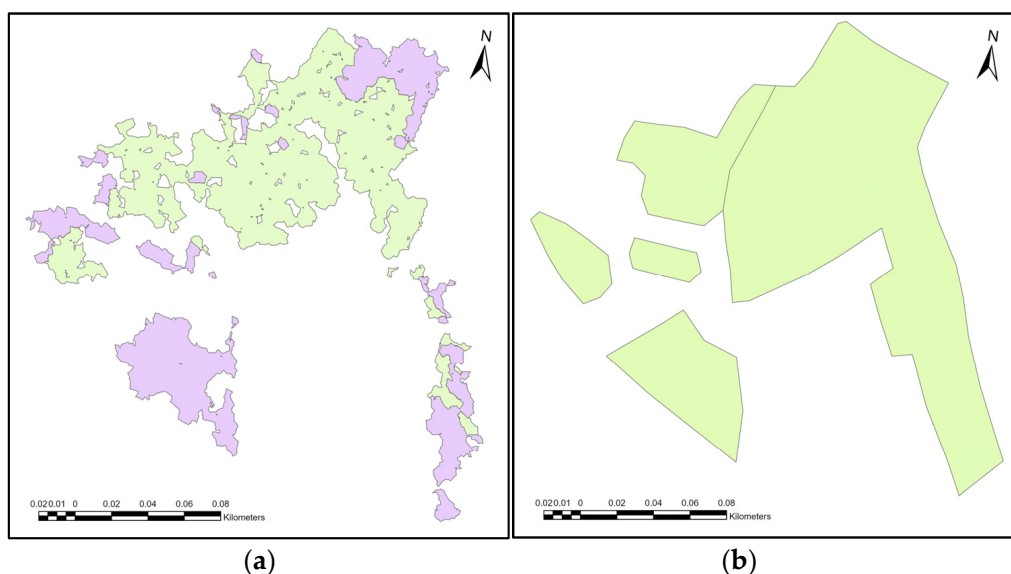

**Figure 2.** (**a**) Shape of the clear-cut area identified using ALS; (**b**) shape of the clear-cut area identified using the forest management plan. The green areas are the year 2020 and the purple areas are year 2021.

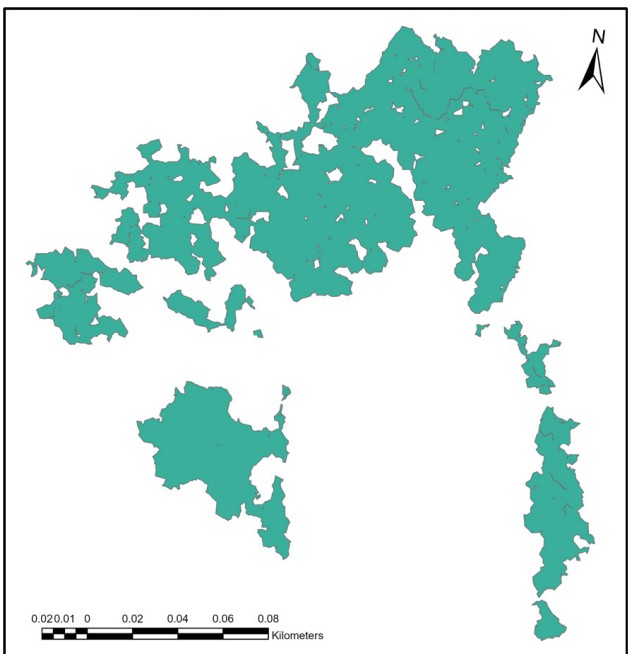

**Figure 3.** Clear-cut area classified using ALS data.

We closed these parts using the "eliminate polygon parts" tool and used the merged 2020 and 2021 clear-cut areas as an input layer. For the "condition", we selected an area that removed parts smaller than the specified value; in our case, the threshold value was 4000 m$^2$ and was selected based on the rule that units below 0.4 ha are not distinguished. After these steps, the polygons were ready for simplification.

The "simplify polygons" tool was used to remove multiple polygon curvatures and simplify shapes. This tool uses four algorithms to simplify polygons as follows:

1. The retain critical points algorithm (Douglas–Peucker) functions based on the concept of reducing the number of points while preserving those that are crucial for defining the polygon's shape. It iteratively eliminates points by dividing the line segment and repeating the process until no more points can be removed. Initially, it creates a line segment by connecting the first and last points. Next, it identifies the point on the line

segment that is farthest from the straight line connecting the endpoints. If the distance between this point and the straight line is smaller than the specified epsilon value (tolerance), the point is discarded. The algorithm then restarts the process with the remaining points between the endpoints, as proposed by Visvalingam and Whyatt in 1990 [18]. This simplified version of the Douglas–Peucker algorithm is demonstrated graphically in Figure 4.

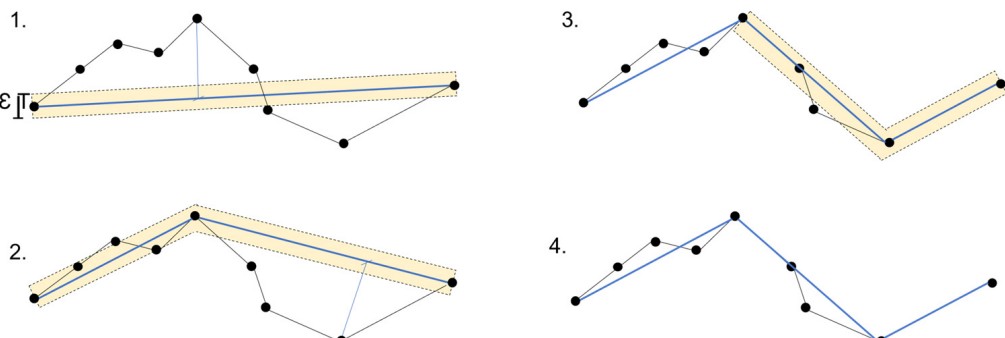

**Figure 4.** The simplification procedure according to the Douglas–Peucker algorithm. In the first step, the first and the last points are connected by a line. In the second step, the algorithm identifies the point that is farthest from the line and then creates a new line originating from that point. In the third step, if the point's distance from the line is less than epsilon, the point is removed. In the fourth step, a new line is created.

2. The Visvalingam–Whyatt algorithm, also known as the retain effective areas algorithm, identifies triangles with effective area and uses that information to remove vertices to simplify the polygon's outline while preserving its overall shape characteristics. This method shares similarities with the Douglas–Peucker algorithm, but instead of a distance-based tolerance, it utilizes a triangle's area as the tolerance criterion. The algorithm starts by identifying the smallest triangle and compares its area to a predefined value also called epsilon [29]. The areas of triangles are continuously compared to the tolerance value. The algorithm removes triangles whose areas are smaller than epsilon. This process is repeated until all triangles with areas smaller than the tolerance value are eliminated [18]. The simplification process using this algorithm is illustrated in Figure 5.

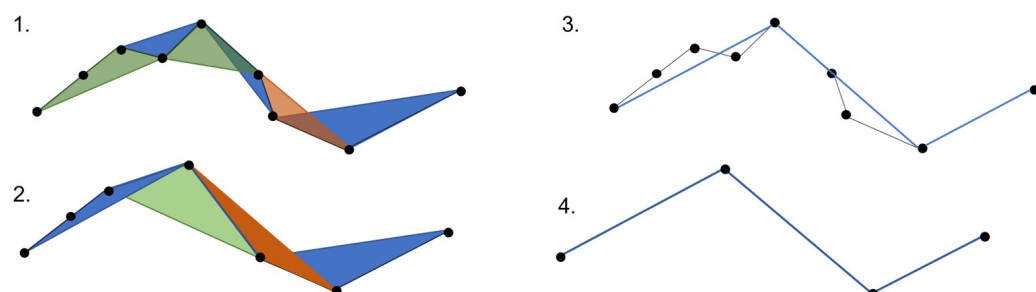

**Figure 5.** The polygon simplification procedure according to the Visvalingam–Whyatt algorithm. In the first step, triangles are formed between the points. In the second step, the smallest triangle is identified, and whether its area is smaller or larger than the specified epsilon is determined. In the third step, if the area is less than epsilon, the point associated with this triangle is discarded. In the fourth step, a new line is created.

3. The Zhou–Jones algorithm (Figure 6), known as the weighted effective area preservation algorithm, assesses the effective areas of triangles associated with each vertex. These effective areas are determined by considering the shape of the triangle and

various metrics, such as flatness, skewness, and convexity [30]. The computation of effective areas for triangles involves applying a weight factor to the initial effective area. This weight factor serves to capture certain aspects of the triangle's shape. Consequently, the introduction of weighted effective area values allows for the distinction between triangles that share the same area but exhibit different shape characteristics. Utilizing various weight definitions enables highlighting of different aspects of triangle shapes. In this context, the functions serve as filters. These filters designate certain triangles as "standard forms" by assigning them a weight of 1, making their effective areas equal under the filter. When examining a triangle's shape characteristics, parameters such the base line length (W), height (H), and length of the middle line (ML) are considered. These parameters allow the measurement of a triangle's flatness, skewness (deviation from an isosceles triangle with the same W and H values), and convexity (orientation relative to a predefined vertex order). There are two models that measure flatness. The first model, which constitutes a high-pass filter, gives priority to taller triangles and reduces the significance of flatter triangles. The second model, a low-pass filter, is identified as a symmetric version of the previously described high-pass filter; its purpose is to eliminate extreme points. The skewness filter is designed to retain points using effective triangles close to being isosceles. The convexity filter is characterized by a constant. If this constant is less than 1, the convexity filter tends to retain points with convex effective triangles. Otherwise, points with concave effective triangles are retained [30]. After weighted areas are calculated, the algorithm strategically eliminates vertices to achieve the maximum possible simplification of the line while still preserving its essential characteristics to the greatest extent possible [31].

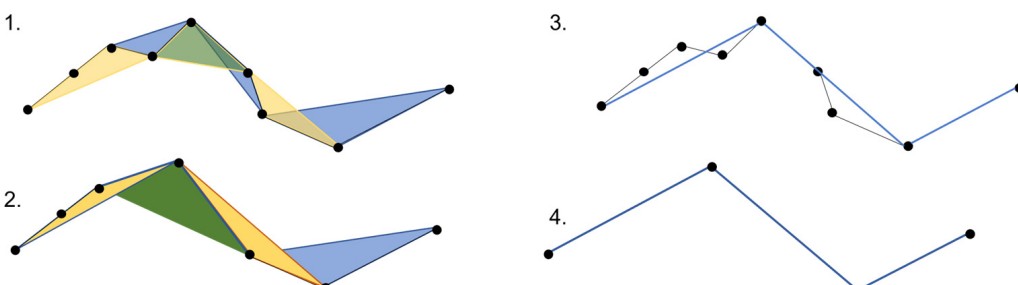

**Figure 6.** The polygon simplification procedure according to the Zhou–Jones algorithm [31]. The algorithm first identifies triangles of effective area for each vertex (1). These triangles are then weighted using a set of metrics to compare the flatness, skewness, and convexity of each area (2). The weighted areas guide the removal of their corresponding vertices to simplify the line while retaining as much character as possible (3). In the last step (4), a new line is created [30].

4.　The retain critical bends algorithm (Wang–Müller) aims to eliminate insignificant bends in polygons. Figures 7–9 depict the process for outline simplification. The minimum diameter for a semicircular bend is set as the tolerance and reference for bend removal. One of the operations in this algorithm is bend elimination (Figure 7); a curved segment is replaced with a straight line. As consecutive straight lines representing bends are not connected, the elimination process must be iteratively performed by removing local minimal bends in each loop. A local minimal bend refers to a bend smaller than both of its neighboring bend points, whereas at the endpoints it is assumed that bends are larger than their neighbors.

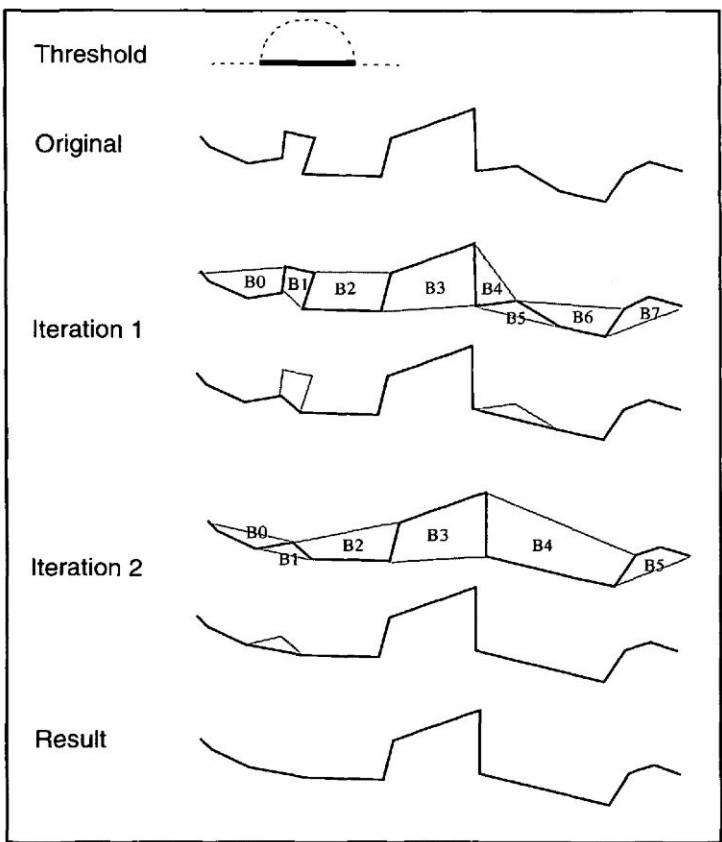

**Figure 7.** Bend elimination by iteration [32]. Numbers B0–B7 are line bends.

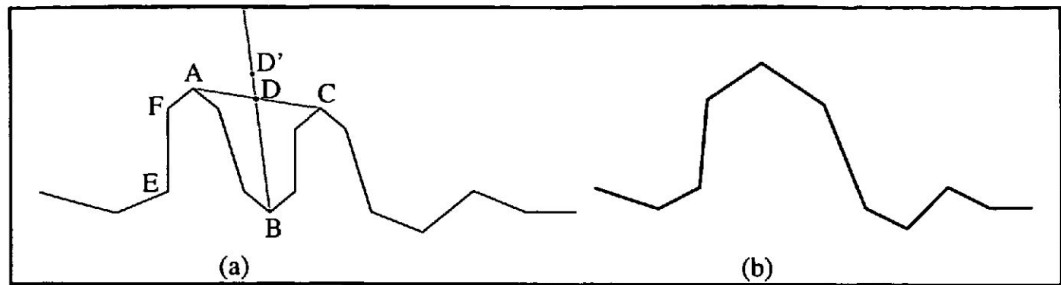

**Figure 8.** Combination of bends [32]. This figure shows three consecutive bends (**a**), and the goal of generalization is to combine the first and the third bends as one (**b**). There are three peaks labelled A, B, and C. Point D is the centre of line AC, and point D′ is the peak of the combined bend.

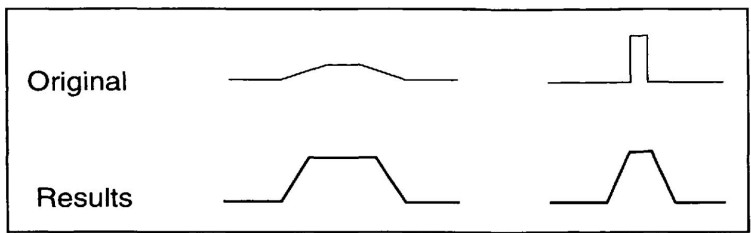

**Figure 9.** Exaggeration using the Gaussian distribution [32].

A second possible operation is bend combination (Figure 8). To determine the bend vertex, distances between the vertices and the two endpoint bend points are calculated, and the vertex with the largest sum is identified as the bend vertex. Subsequently, point D′

is created, representing the midpoint in the bend line, and becomes the new vertex. Finally, the left half of bend 1 and the right half of bend 3 are moved toward the new vertex D'.

The third operation is exaggeration (Figure 9). In this case, shape modification is achieved by enlarging and partially modifying the form. This method uses the Gaussian distribution. During the operation, a central point for translation is found, and instead of moving the endpoints farther from the center, the translation diminishes gradually from the center to the edge [32].

In the "simplify polygon" tool, in addition to establishing the simplification algorithm itself, the so-called simplification tolerance needs to be set. The simplification tolerance parameter is different for each algorithm, and it is referred to later in this text as "parameter" (occasionally it is referred to in the literature as "number"). In the retain critical points (Douglas–Peucker) algorithm, the tolerance parameter refers to the maximum perpendicular distance between each vertex and the resulting simplified line. In the retain critical bends algorithm (Wang–Müller), the tolerance parameter corresponds to the diameter of a circle that approximately represents a significant bend. In the retain weighted effective areas algorithm (Zhou–Jones algorithm), the tolerance square parameter represents the area of a significant triangle formed by three consecutive vertices. The more the triangle deviates from equilateral, the more weight it receives, so it is less likely to be removed. In the retain effective areas (Visvalingam–Whyatt) algorithm, a tolerance square parameter corresponds to the area of a significant triangle formed by three consecutive vertices [31].

Each step was automatically processed using the Python programming code within ArcGIS Pro (Figure 10).

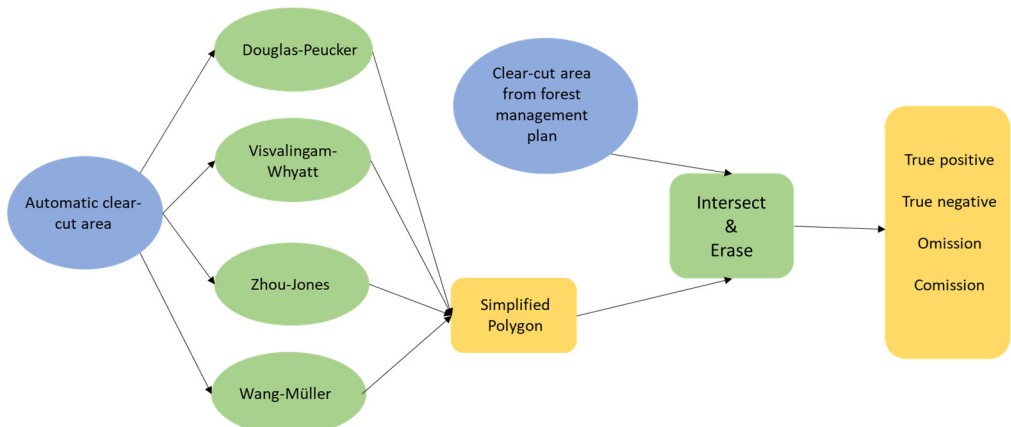

**Figure 10.** The schematic model of data processing in ArcGIS Pro and the creation of evaluation metrics.

### 2.3. Statistical Analysis and Accuracy Assessment

We used error metrics to assess accuracy. We calculated the error of omission and the error of commission.

Classification results were used to estimate accuracy. The confusion matrix was used to evaluate correctly identified clear-cut areas; the matrix provided a summary of two types of errors [25]. Commission errors refer to areas characterized as clear-cut areas by the algorithm that are not actual clear-cut areas. Omission errors refer to actual clear-cut areas not identified as such by the algorithm [33]. Overall accuracy describes how many p pixels (of the total) are classified correctly for all classes [34].

R 4.2.3 software was used to perform the statistical analysis [35]. The accuracy of each algorithm type in relation to the parameter value was compared using generalized additive models (GAMs) in the mgcv package v1.8.42 [36] (Wood, 2017) due to strong nonlinear relationships. GAMs with Gaussian errors were used because the accuracy measurements had a restricted range, and the variance was homoscedastic. We fitted an ANCOVA model with the parameter as a covariate and the algorithm type as a factor. We compared a model

with and without interaction using the AIC. Thin-plate spline was used to fit the nonlinear trend. The resulting model was plotted using the visreg package v2.7.0 [37].

## 3. Results

The analysis showed that the polygon simplification method was suitable for simplifying polygons in forest-clearing detection.

There was a significant interaction between the parameter and algorithm type. Comparison of the accuracies among the four algorithms revealed a significant difference (GAM, $F_3 = 5507$, $p < 0.0001$, $R^2 = 0.98$, Figure 11). For parameter values less than 10, the accuracies of the four algorithms were similar, but for values higher than 10, the Wang–Müller algorithm outperformed the other three algorithms. The Wang–Müller algorithm's maximum accuracy was achieved for a parameter value of 22.

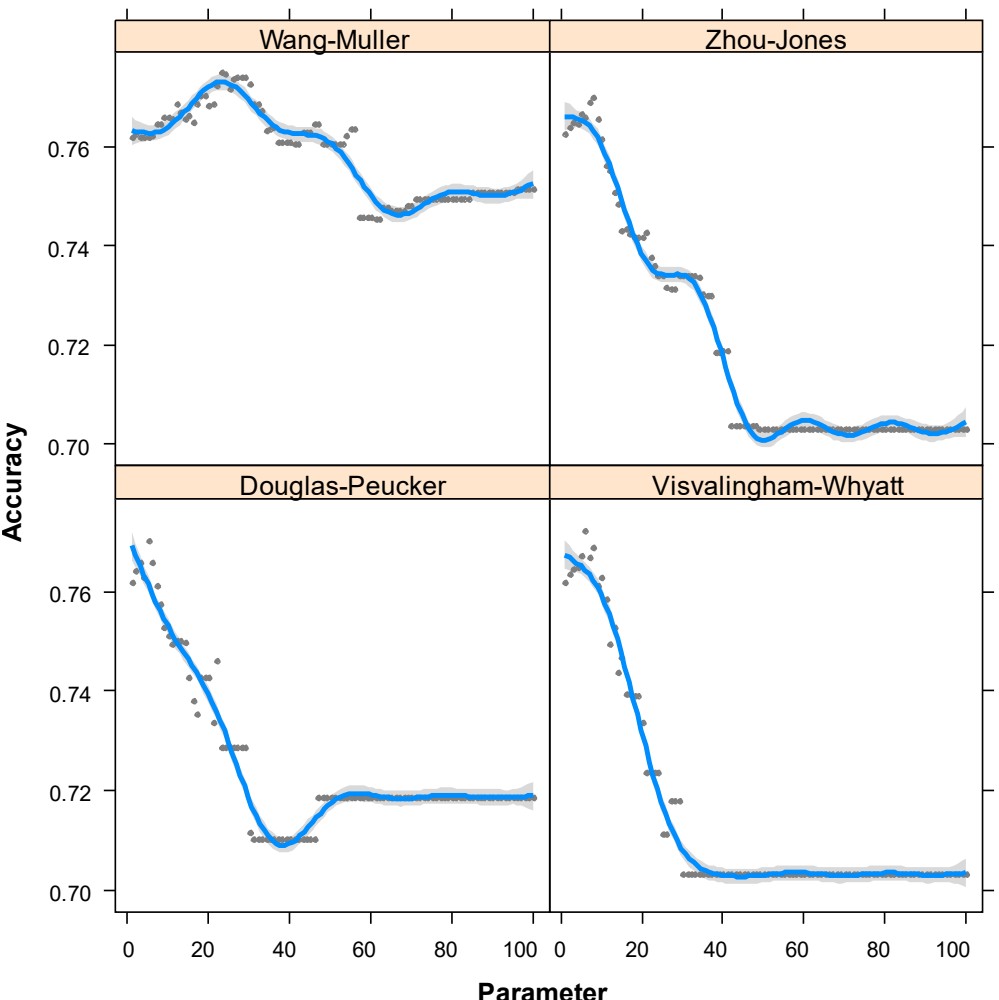

**Figure 11.** The relationship between the parameter values and accuracies for the four algorithm types. The parameters selected were the Douglas–Peucker algorithm's new line, the Wang–Müller algorithm's circle diameter, the Zhou–Jones algorithm's significant triangle area, and the Visvalingam–Whyatt algorithm's significant triangle area. Estimated curves (blue) with their 95% confidence bands (gray) are shown. The parameters were the numerical settings for each algorithm's simplification tolerance.

When comparing commission and omission errors, matrices showed that the Wang–Müller algorithm had the highest commission error but also the smallest omission error (Figure 12).

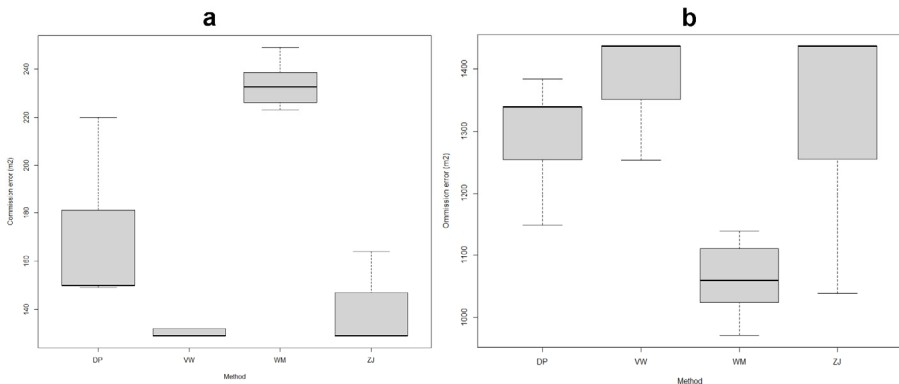

**Figure 12.** (**a**) Commission and (**b**) omission errors in the individual methods.

## 4. Discussion

Cartographic generalization is a crucial phase in the map production process [38]. Our study demonstrated, in a forestry application, the use of simplification algorithms in ArcGIS Pro. Previous studies focused on simplifying roof shapes when using ALS and vector maps to create 3D building models, in which the authors used the Douglas–Peucker algorithm first, followed by the partitioning method, combining step edges with footprint maps [7]. Other articles focused on the simplification of urban residential area plans using raster and vector models with mathematical morphology and pattern recognition assisted by applying a neural network [39]. Using a simplification algorithm was also found to be appropriate when diluting data and preserving the trajectory curve of acquired in-vehicle GPS data, for which the Douglas–Peucker algorithm was found to be suitable [40]. Another article described utilizing generalization algorithms in an ArcGIS environment [41]; these authors used the same algorithms as in our analysis. After using the simplification algorithm, they smoothed the polygons in the GIS environment. Their results showed that the Douglas–Peucker algorithm was suitable for data compression and the removal of redundant polygon details. The disadvantage of this algorithm is that the resulting line contains sharp angles and spikes. Compared with the Douglas–Peucker algorithm, the Wang–Müller algorithm prioritizes the input geometry to a greater extent, which requires additional processing time [42]. Their article describes the development of a new algorithm to simplify polygons and lines representing hydrographic lakes and streams. To assess their new algorithm, they compared it with the well-known Douglas–Peucker algorithm and the Wang–Müller bend simplification algorithm. Their algorithm has no user-defined parameters, and it defines an error band that does not allow the simplified line to cross it. This ensures the accuracy of the resulting line. We also used the Douglas–Peucker and Wang–Müller algorithms, and ArcGIS uses two additional algorithms, the Zhou–Jones and Visvalingam–Whyatt algorithms, as described above. Previous studies compared the Douglas–Peucker and Wang–Müller algorithms, showing that the point-remove (Douglas–Peucker) algorithm could remove more points from the line, resulting in a more streamlined and adaptable database for users. However, it also led to the loss of the original line's topological characteristics. The bend-simplify (Wang–Müller) algorithm removed fewer points from the line, and it preserved a topology closer to that of the original line. When combined with other topological elements on a map to assess proximity and adjacency, the Wang–Müller algorithm demonstrated better adjustment [43].

Modification of the clear-cut areas created by airborne laser scanning is essential for cartographic display. Manual processing is tedious and does not provide any significant advantages in cases with a large number of clear-cut polygons. In our study, 29 polygons had, in summary, 16,649 ALS vertices that required adjustment, leading to 841 plotted vertices that were illustrated in the forest management plan. Implementing the simplification algorithm enhanced the efficiency of polygon editing; manual modification is as much as two orders of magnitude slower than using this simplification tool.

## 5. Conclusions

Most of the published literature to date addresses the simplification of polygons in cartography, focusing on simplifying lines or polygons that represent watercourses, buildings, or land-unit boundaries. In this study, we addressed the problem of excessive detail displayed in forest clear-cut areas that occurs after autodetection data are extracted from airborne laser scanning. Polygon simplification is a suitable method to simplify the shapes of clear-cut areas when creating forest base maps. We used the ArcGIS Pro "geoprocessing" tool to simplify polygons in order to estimate and compare the accuracy of individual algorithms when compared with ground-truth data from the forest management plan. This tool used four simplification algorithm parameters: retain critical points (Douglas–Peucker algorithm), retain critical bends (Wang–Müller algorithm), retain weighted effective areas (Zhou–Jones algorithm), and retain effective areas (Visvalingam–Whyatt algorithm). Our results show that the Wang–Müller algorithm performed best when using a parameter in the 20 to 25 m range.

**Author Contributions:** Conceptualization, Z.M. and P.S.; methodology, Z.M., S.P. and P.S.; software, Z.M. and P.S.; validation, Z.M., S.P. and P.S.; formal analysis, Z.M.; investigation, Z.M.; resources, P.S.; data curation, Z.M. and P.S.; writing—original draft preparation, Z.M. and P.S.; writing—review and editing, Z.M. and P.S.; visualization, Z.M.; supervision, P.S.; project administration, P.S.; funding acquisition, P.S. All authors have read and agreed to the published version of the manuscript.

**Funding:** This research was funded by the Ministry of Agriculture of the Czech Republic, grant number QK21010435.

**Data Availability Statement:** Data are available on request for research purposes.

**Conflicts of Interest:** The authors declare no conflict of interest.

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
