# Peer review of "Benchmark for Automatic Clear-Cut Morphology Detection Methods Derived from Airborne Lidar Data"

_forests, doi:10.3390/f14122408_

Round 1
Reviewer 1 Report
Comments and Suggestions for Authors
Melichova et al.: Benchmark for methods for automatic clear-cut morphology detection derived from airborne lidar data
The manuscript addresses the problem of identifying and estimating clear-cut
areas in ALS-based forest-area datasets. Four different algorithms for
polygon simplification to denoise the data are compared with each other
and their accuracy is assessed, with a forst management plan serving
as ground-truth.
The topic of polygon simplification is of high relevance in geoinformatics
and spatial data analysis, and all four considered algorithms are well-known
and often-used. The comparison carried out in this manuscript, which uses
statistical tools, is thus interesting and of great significance in
forest-related geodata evaluation. The techniques used in this study
seem to be sound and well-founded.
However, the presentation of the methods and results in the paper has
still considerable deficiencies:
(1) The description of the third algorithm which is used in the study
(Zhou-Jones) (lines 197-203) is vague and unprecise, compared with
the other three employed algorithms. It should be explained in more detail
(e.g., "various metrics like flatness, skewness, and convexity": are there
even more metrics? How are they quantified?) Furthermore, the corresponding
Fig. 6 lacks quality (the pixels are visible) and does not indicate
the way the simplification is obtained, unlike the illustrations for
the other three algorithms.
(2) Fig. 10, showing a diagram of the dataflow, is confusing. As far
as I understand from the text, the step "Simplify Polygon" (yellow
in the diagram) uses alternatively one of the four algorithms, with
"Douglas-Peucker" being one of them. So, why is "Douglas-Peucker"
occurring once again in the diagram (in the green objects) - and not
the three other algorithms? Is it called twice, or does the arrow
from "Simplify Polygon" to "D.-P." mean that D.-P. is called by
Simplify Polygon? Or is there an additional workstep, with D.-P.
playing a preferred role in this second step? Finally, shouldn't the
labels for both lower green objects read "Douglas-Peucker commission
error" and "Douglas-Peucker omission error" (with "error" missing
in the diagram)?
(3) It is not defined what "the Number" means in lines 274, 286, 288,
and "num" as the axis label in Fig. 11. Later on, in the legend of
Fig. 12, there is an explanation which seems to tell that this "number"
is identical with the respective parameters for "tolerance" of the four
algorithms (cf. lines 234-238 before). The authors should use the same
words for the same things throughout the paper in order to avoid confusion.
(4) Fig. 11 seems to be a simplified version of Fig. 12. However, this
is not explained in the text. Furthermore, the meaning of the colored
markers on the x axis is unclear. - Does Fig. 11 show anything which
is not shown in Fig. 12? Otherwise it can be omitted.
(5) The diagrams in Fig. 13 show cryptic axis labels ("F$Sp" etc.),
and are lacking a unit for the numbers at the y axis.
(6) Lines 307 and 309 seem to refer to the same diagram (a) in Fig. 14,
but the figure legend tells that accuracy is displayed in part (b),
and the value for Wang-Mueller is here not 0.76 but something near 0.53.
Furthermore, the y-axis labels in the diagrams in Fig. 14 contradict
the description of (a) and (b) in the figure legend.
(7) Line 139: It should be "Fig. 2" here instead of "the previous picture".
(8) The number of the figure on p. 4 (line 152) should be "Figure 2"
instead of "Figure 1".
(9) Line 167: The number "400" lacks a unit.
(10) Line 278: "number of ..": Something is missing here.
(11) Important: The whole text needs a thorough review and correction
of the used English language, preferentially by a native English speaker!
Many sentences are really hard to understand because of language flaws.
see Comments and Suggestions for Authors
Author Response
Dear reviewer 1, thank you very much for your comments and suggestions to improve the article. We agreed to all of them and you can find in the attached document.

Reviewer 2 Report
Comments and Suggestions for Authors
Dear authors,
the contribution sounds interesting and scientific in general. However, some parts of the paper need to be revised. Below, you can find some suggestions that in my opinion will help improve the paper.
In the introduction section: Please, highlight and describe your contribution and the novelty of your work.
Line 131: please, describe at least briefly the filtration and the classification process. It is important in terms of results.
all the figs: please, increase the quality of the graphical representation.
Line 152 it should be fig 2
Fig4: add a short description of the steps into the legend, as well as for Fig 5
Author Response
Dear reviewer, thank you very much for your comments and suggestions to improve the article. We agreed and incorporated all of them. Please see attached document.
